# Marrying chemistry with biology by combining on-chip solution-based combinatorial synthesis and cellular screening

Maximilian Benz [1], Mijanur R. Molla[1,2], Alexander Böser[1], Alisa Rosenfeld[1] & Pavel A. Levkin [1,3]

Drug development often relies on high-throughput cell-based screening of large compound libraries. However, the lack of miniaturized and parallelized methodologies in chemistry as well as strict separation and incompatibility of the synthesis of bioactive compounds from their biological screenings makes this process expensive and inefficient. Here, we demonstrate an on-chip platform that combines solution-based synthesis of compound libraries with high-throughput biological screenings (chemBIOS). The chemBIOS platform is compatible with both organic solvents required for the synthesis and aqueous solutions necessary for biological screenings. We use the chemBIOS platform to perform 75 parallel, three-component reactions to synthesize a library of lipidoids, followed by characterization via MALDI-MS, on-chip formation of lipoplexes, and on-chip cell screening. The entire process from the library synthesis to cell screening takes only 3 days and about 1 mL of total solutions, demonstrating the potential of the chemBIOS technology to increase efficiency and accelerate screenings and drug development.

---

[1] Karlsruhe Institute of Technology (KIT), Institute of Toxicology and Genetics (ITG), Hermann-von-Helmholtz-Platz 1, 76344 Eggenstein-Leopoldshafen, Germany. [2] Department of Chemistry, University of Calcutta, 92, APC Road, Kolkata 700009, India. [3] Karlsruhe Institute of Technology (KIT), Institute of Organic Chemistry (IOC), Kaiserstraße 12, 76131 Karlsruhe, Germany. Correspondence and requests for materials should be addressed to P.A.L. (email: levkin@kit.edu)

The pharmaceutical industry struggles to meet the ever-increasing demand for new drugs. A decrease in the development of new drugs has been observed for years[1]. From 1991 to 2000, the total number of new drugs discovered in the 21 leading countries was 367, dropping to 251 for the period 2001–2010[2]. Fewer and fewer new drugs are being tested in clinical trials, and R&D takes longer to develop potential new drug candidates[1].

The process of developing drugs as well as various other fundamental and applied biological experiments begins from the organic synthesis of compound libraries, followed by their high-throughput screening in biological assays to identify few active molecules (hits) (Fig. 1). Most of the compounds available in primary and secondary libraries are synthesized individually via standard organic synthesis usually involving large quantities of reagents and organic solvents. This makes library synthesis an extremely lengthy and costly process taking many years and consuming valuable resources. Despite these well-established procedures and methods, it often takes over 20 years and $2–4 billion between a single drug's first synthesis[3], biological screening[4,5] and ultimate approval[6].

There are many reasons for such slow progress in drug development, ranging from typical problems such as poor lead optimization, limited specificity, and potential toxicity. Furthermore, due to high costs and low availability, primary screenings are usually only done once (without repeating or varying concentrations), and only in big pharmaceutical companies or screening centers. Organic synthesis in flasks with large volumes, organic solvents, and harsh conditions renders it incompatible with any biological assay, which are usually performed in polystyrene microtiter plates, requiring small volumes, mild aqueous conditions, and compatibility with the corresponding infrastructure (microscopy, pipetting, parallelization, and miniaturization).

Microarrays have been used to try to overcome these problems. Solid-phase synthesis (SPS) in the microarray format is one example showing the potential of miniaturization and combination with the biological part. Microarrays of DNA, peptide microarrays, small molecule microarrays or oligosaccharide microarrays produced by SPS have been demonstrated[7–13]. Although these SPS methods have accelerated the development and investigation of large and diverse compound libraries because

of their inherent miniaturization and parallelization, solution-based synthesis offers a much broader scope of chemical reactions. Moreover, most of the existing SPS methods are incompatible with more physiologically relevant cellular assays (2D or 3D cell culture) where freely diffusing compounds and therefore compartmentalization of individual cell experiments are required.

Surprisingly, despite the well-established procedure for high-throughput screenings (HTS) and drug discovery, and apart from a pharmaceutical industry that reportedly screens millions of compounds every year, there are very few academic publications on HTS entailing more than 10,000 compounds. For example, Torrance et al. screened 29,440 different compounds to identify active agents that inhibit tumor xenografts[14]. The reasons for the above are clearly understood if we approximate the costs of such HTSs starting from the synthesis part and up to biological screenings. For example, if we had to synthesize 100,000 different compounds for a subsequent biological screening, even if each reaction required only 10 mL of solvents and biological assay (standard 384-well MTPs) would need 60 µL (20 µL in triplicates), the total volume of organic solvents would amount to 1000 L and aqueous solutions at least to 6 L for each bioassay. The most important limitation, however, is the time needed for the synthesis part. For example, if one chemist had to do this library synthesis at a speed of five compounds per day, it would take 55 years. This illustrates the underlying problem for the entire field of drug discovery and explains the slow development and exorbitant costs (on average $2–4 billion per drug[15–18]) in this field essential to everyone.

Although it remains very challenging to unify chemical synthesis with the biological screening part to enable faster transfer and utilization of the synthesized compounds in biological screenings, we believe this is the most efficient pathway to solve the aforementioned problems.

Here we develop a microarray platform compatible with the solution-based organic synthesis of combinatorial libraries of small molecules that are easily screenable in various biological and cell-based assays using the same platform (chemBIOS; Fig. 1). As a proof-of-concept, we use the chemBIOS platform to synthesize a library of lipid-like molecules (lipidoids), followed by the on-chip preparation of lipoplexes and the on-chip cell screening to identify transfection reagents.

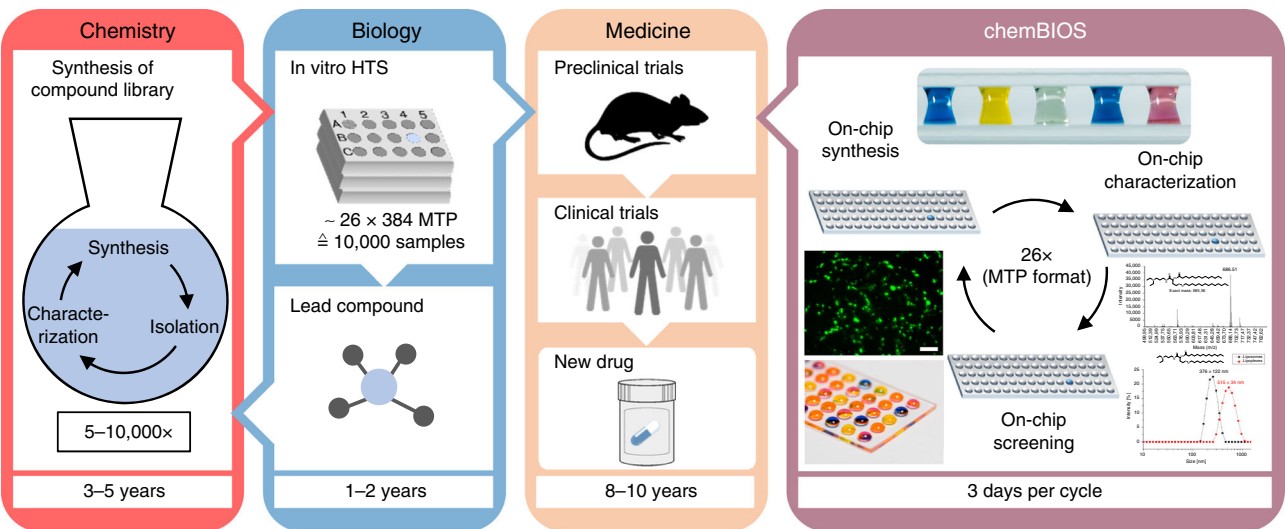

**Fig. 1** Schematic describing the process of drug discovery. Strict separation between chemistry (synthesis) and biology (assays) makes the process of drug discovery inefficient. ChemBIOS unifies miniaturized solution-based chemical synthesis performed in a microarray format, characterization and biological screening, and thus, all steps in early-stage drug discovery take a few days rather than years. Scale bar: 200 µm

## Results

**Manufacturing and characterization of the platform.** Our entire validation of the chemBIOS process was conducted in four steps: (i) manufacturing and characterizing the platform, (ii) on-chip synthesis and characterization of a lipidoid library, (iii) on-chip formation of liposomes and lipoplexes, and (iv) on-chip cellular screening of produced lipoplexes.

A compound library was synthesized using omniphilic-omniphobic microarrays prepared on glass slides compatible with low surface tension organic liquids, thus called Low Surface Tension Liquids (LSTL) slides[19]. To manufacture LSTL slides, we first modified the surface of the glass slide with chloro (dimethyl)vinylsilane to produce a monolayer of reactive vinyl groups on the surface. Afterwards, the photochemical patterning of the surface via the thiol-ene photoclick reaction (Fig. 2a) led to the formation of omniphilic spots functionalized with cysteamine hydrochloride ($\theta_{adv}$(DMSO) = 32.9 ± 2.4°, $\theta_{rec}$(DMSO) = 17.9 ± 2.2°), spatially separated by omniphobic borders functionalized with 1H, 1H, 2H, 2H-perfluorodecanethiol (PFDT) ($\theta_{adv}$(DMSO) = 87.0 ± 3.5°, $\theta_{rec}$(DMSO) = 72.2 ± 1.7°) (Fig. 2c and Supplementary Fig. 1). In all, ± values of all contact angle measurements are standard deviations based on triplicate experiments. Due to the omniphilic-omniphobic properties of these patterns and excellent dewetting characteristics of the fluorinated regions, low surface tension liquids including various common organic solvents (such as n-hexane, ethanol, 1-decanol, DMF, and DMSO (Supplementary Fig. 2)) could form arrays of microdroplets via the effect of discontinuous dewetting (Fig. 2e)[19]. The shape and volume of these droplets could be defined by varying the geometry of the patterns. Depending on the patterns and solvent, we could create droplets from few picoliters up to microliters[20]. However, LSTL slides cannot be used to form droplet arrays employing aqueous solutions with much higher surface tension, due to the relatively high receding contact angle of the water on the hydrophilic regions ($\theta_{rec}$(H$_2$O) = 33.7 ± 0.8°; Supplementary Fig. 1).

Therefore, in the next step we produced High Surface Tension Liquids (HSTL) slides—glass slides coated with a porous polymer layer patterned with hydrophilic spots separated by hydrophobic barriers, leading to the effect of discontinuous dewetting of aqueous solutions[21,22]. The preparation of HSTL slides occurs via the photochemical patterning of a thin, porous layer of poly(2-hydroxyethyl methacrylate-co-ethylene dimethacrylate) (HEMA-co-EDMA polymer). Highly hydrophilic spots functionalized with 2-mercaptoethanol ($\theta_{adv}$(H$_2$O) = 19.2 ± 1.6°, $\theta_{rec}$(H$_2$O) = 6.3 ± 1.1°) and spatially separated by highly hydrophobic borders functionalized with PFDT ($\theta_{adv}$(H$_2$O) = 159.3 ± 6.8°, $\theta_{rec}$(H$_2$O) = 139.6 ± 2.2°) were generated via the photochemical thiol-yne photoclick reaction (Fig. 2b, d and Supplementary Fig. 1). The combination of highly hydrophilic spots possessing very strong affinity to aqueous solutions and hydrophobic, extremely water-repellent regions enabled the formation of arrays of droplets of high surface tension aqueous solutions such as liposomes or cell suspension using discontinuous dewetting (Supplementary Fig. 3 and Supplementary Movie 1).

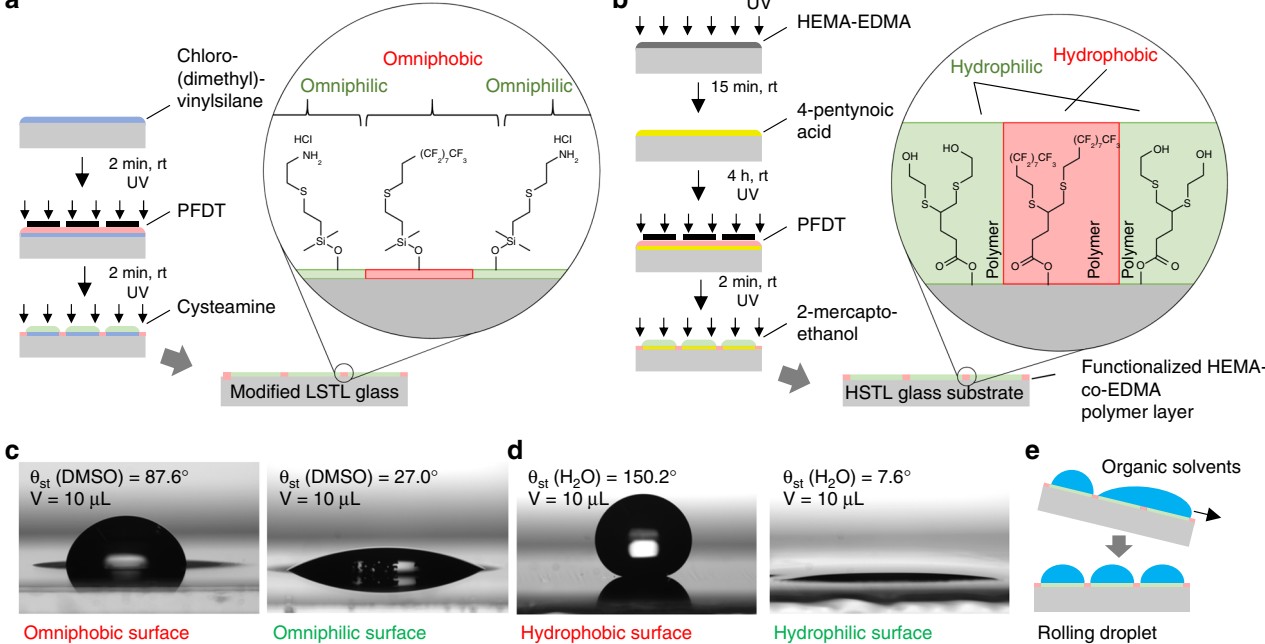

**Fig. 2** Manufacture and characterization of patterned slides used for the chemBIOS platform. The chemBIOS platform consists of two types of patterned glass slides. **a** Slides compatible with Low Surface Tension Liquids (LSTL slides) are produced by silanizing the glass surface with chloro(dimethyl) vinylsilane and patterning via the photochemical thiol-ene click reaction. Omniphobic borders are generated by a reaction with perfluorodecanethiol (PFDT), followed by the formation of omniphilic spots by cysteamine hydrochloride. Spot diameter 2.83 mm; hydrophobic borders width 1.67 mm. **b** Slides for High Surface Tension Liquids (HSTL slides) are manufactured via a polymerization reaction to apply a porous polymer layer of poly(2-hydroxyethyl methacrylate-co-ethylene dimethacrylate). Functionalization with 4-pentynoic acid enables further surface patterning by thiol-yne photoclick chemistry. Hydrophobic borders are generated by PFDT, followed by the formation of hydrophilic spots using 2-mercaptoethanol. Spot diameter 2.83 mm; hydrophobic borders width 1.67 mm. **c** Photographs of droplets of DMSO on the omniphobic surface and omniphilic surfaces used for the LSTL patterns with corresponding static contact angle. Droplet volume: 10 µL. **d** Photographs of droplets of water on hydrophobic and hydrophilic surfaces used for the HSTL slides with corresponding static contact angle. Droplet volume: 10 µL. **e** Schematic showing the effect of discontinuous dewetting on patterned LSTL slides, which enables manual generation of organic droplet arrays

**On-chip synthesis and characterization of a lipidoid library.**
We then used the LSTL slides to develop and optimize the parallel, combinatorial synthesis of cationic lipid-like molecules (lipidoids) to be used in the following cell screening experiment. For this purpose, we chose a one-pot three-component reaction based on thiolactone opening by an amine, followed by a disulfide exchange reaction (Fig. 3b)[23]. We used two methods to apply solutions in droplet-array format to the slides: a printing method, where we applied droplets to each spot via a non-contact liquid dispenser system, and the rolling droplet method, where we applied solutions via discontinuous dewetting. The rolling droplet method was only used to synthesize the lipidoids characterized by MALDI-TOF mass spectrometric analysis and is described below. Solutions of various amines (A1–A5) in DMSO were applied in a column-by-column manner on an LSTL slide A to form an array of different amines in each row (Fig. 3a). The thiolactone and pyridyl disulfide reactants were premixed together in DMSO in different combinations and applied on a LSTL slide B in row-by-

row manner perpendicular to the columns in slide A (Fig. 3a). By sandwiching both slides using an aligner device (Supplementary Fig 4), the droplets on slides A and B were merged in array format, initiating the chemical reaction simultaneously in each droplet on the array (Fig. 3a). Using this method, we were able to synthesize a library of 25 different compounds, each in triplicate on the same slide in a single step (Supplementary Table 1). The reaction was carried out for 2 h at room temperature.

We used the released 2-thiopyridone to monitor the reaction kinetics and estimate the yield via UV-Vis spectroscopy. The extinction coefficient of 2-thiopyridon in DMSO/acetonitrile (1:10) was independently estimated to be $2040\,M^{-1}\,cm^{-1}$ at 370 nm (Supplementary Fig. 5a). To monitor the progress of the reaction, we took one droplet from the array, diluted it in 90 µL acetonitrile, and analyzed it by UV-Vis. Based on those UV-Vis results, the reaction was complete after two hours (Supplementary Fig. 5b). By quantifying the absolute concentration of 2-thiopyridone, we indirectly calculated the average yield of 89 ± 15% by Beer-Lambert

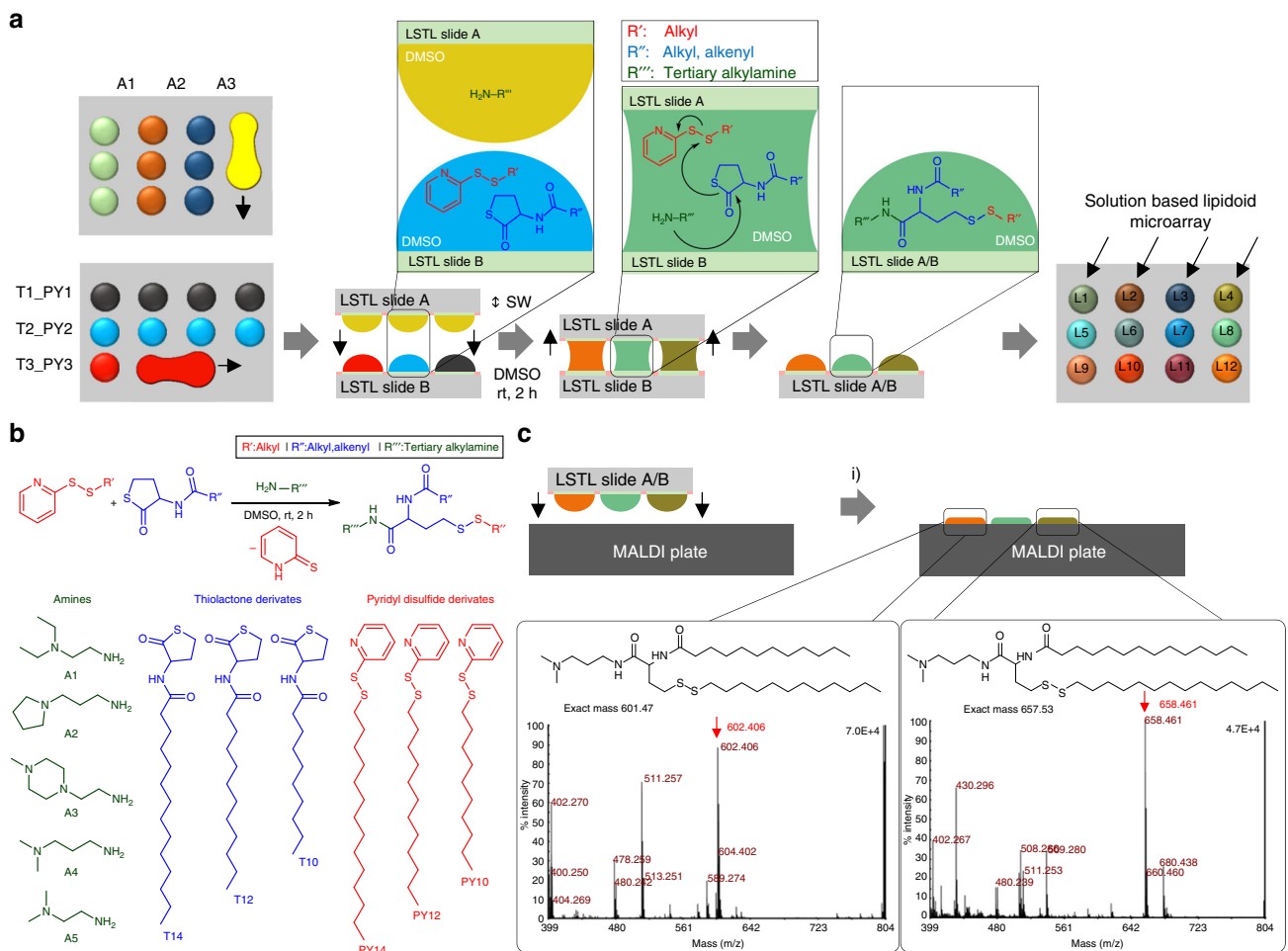

**Fig. 3** On-chip synthesis and characterization of a lipidoid library. **a** Schematic description of combinatorial synthesis of lipidoids using the chemBIOS platform. In the first step, various amines dissolved in DMSO are applied column-by-column onto a low surface tension liquids (LSTL) slide A. Then, mixtures of thiolactone and pyridyl disulfide derivates dissolved in DMSO are applied on a LSTL slide B in rows oriented orthogonally to the columns on slide A. The application is done by rolling a droplet of solution to form rows of separate droplets. In the next step, slide A and slide B are sandwiched (SW) with each other to bring individual droplets into contact to merge the droplets, thereby initiating the reaction. The reaction was completed after 2 h at room temperature. 25 different lipidoids per slide were synthesized in triplicates on a single slide. **b** Scheme of the reaction and a list of all different precursors for the reaction. **c** To characterize produced compounds, slide A was sandwiched onto a MALDI plate, followed by the addition of matrix and drying the plate. MALDI-TOF mass spectrometry measurement (examples of spectra shown in the insets, positive mode) enables rapid and convenient characterization of the entire library of synthesized compounds. (i) sandwiching, followed by adding 10 mg mL$^{-1}$ α-cyano-4-hydroxycinnamic acid solution of 1:1 acetonitrile: water containing 0.1% v/v trifluoroacetic acid. Source data are provided as a Source Data file

law (Supplementary Fig. 5c and Supplementary Table 1). In all, ± values of all reaction yields are standard deviations based on triplicate experiments. The products of the reaction were analyzed via MALDI-TOF mass spectrometry (MS). The open system and flat substrate of chemBIOS enabled us to transfer and copy the entire compound library in a single step by stamping it onto a MALDI plate (Fig. 3c and Supplementary Fig. 6). Moreover, we characterized the raw product by [1]H-NMR (Supplementary Fig. 7) and on-chip ATR-IR spectroscopy (Supplementary Fig. 8).

The open format of the chemBIOS system is compatible with further on-chip steps such as investigating, treating or converting the chemical compound library, to achieve high-throughput parallel on-chip purification. In a proof-of-principle experiment, we showed how crude products could be purified by on-chip two-phase liquid extraction (Fig. 4a, b). Therefore, we dissolved Nile red and methylene blue in 1-octanol which resulted in a dark blue solution. Next, we applied this solution on several spots on an LSTL slide D and sandwiched that slide with an HSTL slide E containing water droplets, resulting in the formation of an octanol-water interface for each droplet (Fig. 4a). Methylene blue is highly water-soluble, and was therefore extracted from the organic phase into the aqueous phase, while water-insoluble Nile red remained in the organic phase. The organic phase turned red after 10 min, while the aqueous phase turned blue, indicating successful separation. We analyzed the mixture before purification, as well as the organic and aqueous phases after purification by UV-Vis spectroscopy (Fig. 4b) and liquid chromatography mass spectrometry (LC-MS) (Supplementary Fig. 9 and 10). Both analytical methods yielded similar results. Using UV-Vis spectroscopy, the concentration of methylene blue measured $11.9 \pm 0.5$ mM in the original organic phase mixture, and $3.3 \pm 0.6$ mM and $4.1 \pm 0.4$ mM in the organic and aqueous phase, respectively, after extraction. The concentration of Nile red was measured 6.0 mM in the original mixture, and $4.3 \pm 0.3$ mM and $0.3 \pm 0.04$ mM in the organic and aqueous phase after extraction. This corresponds to the purification of the methylene blue from 66% in the mixture to 91% in the aqueous phase and that of Nile red from 34% in the mixture to 57% in the organic phase via just one purification step. In all, ±values of all concentration measurements are standard deviations based on triplicate experiments.

**On-chip formation of liposomes and lipoplexes.** In the next step, we transferred the lipidoid library from the LSTL slide A to an HSTL slide C (Fig. 5a). This step was performed to transfer lipidoids into aqueous cell-compatible droplets simultaneously, forming either liposomes or lipoplexes required for cell transfection experiments. To make lipoplexes or liposomes, the LSTL slide A containing dried lipidoids was sandwiched with HSTL slide C containing an array of droplets of an aqueous sodium acetate buffer (50 mM, pH 5) with or without plasmid pCS2-GFP (75 ng μL$^{-1}$), respectively. The buffer also contained 0.04% w/v gelatin, 3.4% w/v sucrose, 0.002% w/v human fibronectin required for the subsequent reverse cell transfection experiment[24]. The sandwiched slides were incubated at 50 °C for 1.5 h to support the formation of liposomes/lipoplexes. As a result, the synthesis products were transferred to the HSTL slide C. Having been transferred into an aqueous environment, the amphiphilic lipidoids spontaneously formed liposomes, subsequently complexing plasmid DNA to form lipoplexes (Fig. 5a). We characterized both liposome and lipoplex solutions by pipetting them directly from the HSTL slide C into a cuvette to analyze via dynamic light scattering (DLS) and zeta potential analysis. The lipoplex solutions revealed significantly larger particles than the corresponding liposome solutions (Fig. 5b). Furthermore, lipoplex solutions demonstrated a lower zeta potential than the corresponding liposomes, indicating the formation of complexes between the positively charged lipidoids and negatively charged DNA (Fig. 5b).

**On-chip cell-based screening of produced lipoplexes.** Next, we investigated the lipoplex library in an on-chip cell-based screening using reverse transfection of HEK293T cells (Fig. 6a).

ScreenFect A was used as a positive control, while dilution buffer without pDNA and untreated cells served as negative controls. The transfection experiment was performed by printing 5 μL of a suspension of $6 \times 10^5$ HEK293T cells mL$^{-1}$ into each spot on the dried lipoplex array slide using a non-contact liquid

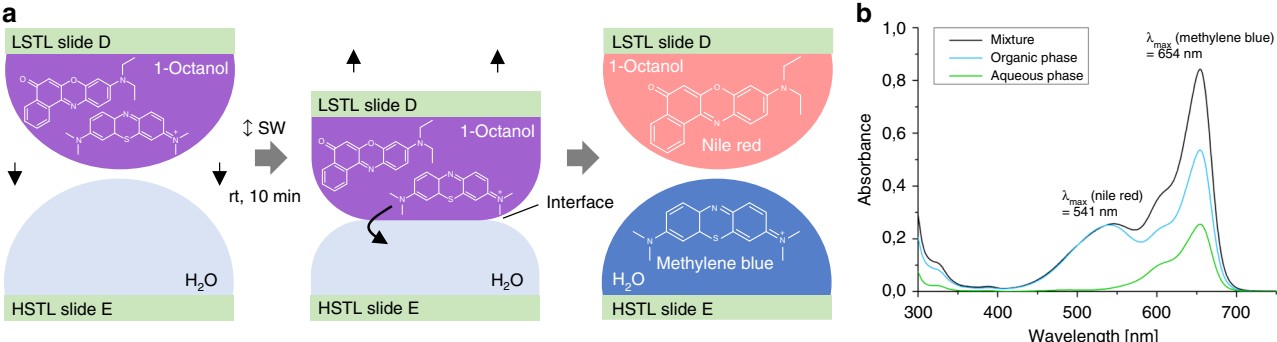

**Fig. 4** On-chip parallel liquid-liquid extraction. **a** Schematic description of the on-chip two-phase liquid extraction process. A mixture of oleophilic Nile red and hydrophilic methylene blue in 1-octanol on an LSTL slide D was separated by sandwiching the slide with an HSTL slide E carrying water droplets for 10 min. An interface between the two non-miscible solvents was formed, and the water-soluble methylene blue was extracted into the aqueous phase. **b** Validation of the on-chip extraction via UV-Vis spectroscopy. The UV-Vis spectrum of the mixture reveals two local absorbance maxima: one at 541 nm corresponding to Nile red, and another at 654 nm corresponding to methylene blue. After the extraction, the intensity of the absorbance maximum of methylene blue in the organic phase decreased while the intensity of Nile red remained constant. In the aqueous phase, no Nile red could be detected. Source data are provided as a Source Data file

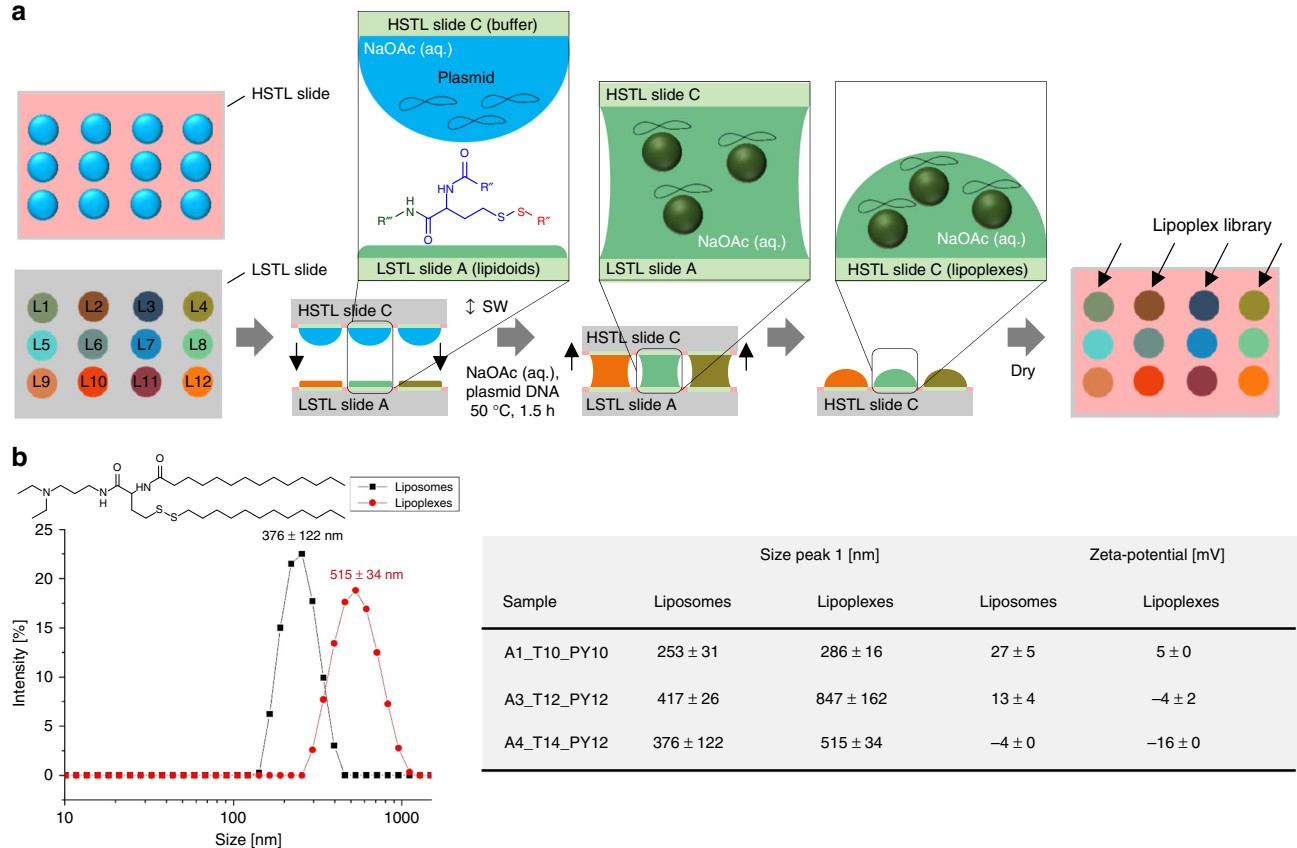

**Fig. 5** On-chip parallel formation of a library of liposomes or lipoplexes. **a** Schematic description of lipoplex formation. An array (HSTL slide C) of aqueous sodium acetate buffer droplets containing sucrose, gelatin, fibronectin and pDNA (pCS2-GFP) was sandwiched with LSTL slide A containing a library of dried lipidoids, followed by incubation at 50 °C for 1.5 h and drying before using the slide C for the following reverse cell transfection experiment. Liposomes were produced in the same way without adding plasmid DNA. **b** Results of dynamic light scattering (DLS) and zeta potential analyses of lipoplexes and corresponding liposomes. Liposomes display smaller particles and higher zeta potential than corresponding lipoplexes.+/− values are standard deviations, n = 3 (number of replicates). Source data are provided as a Source Data file

dispenser. After 48 h of cultivating and staining the cells with Hoechst and propidium iodide (PI), we determined the transfection efficiency of each sample of the array by fluorescence microscopy (Fig. 6b, c and Supplementary Table 2). We observed that sample A3_T14_PY14 was the best performing transfection agent in our screening with 52 ± 4% average transfection efficiency (Fig. 6b). Viability of cells in the negative control experiment (spots without lipidoids and only treated with buffer solution) was 97 ± 1%, while cells transfected with the most efficient lipoplexes demonstrated ~34% viability as measured by PI/Höchst staining (Supplementary Fig. 8).

## Discussion

The process of developing biologically active compounds and potential drug candidates is difficult due to several factors. The transfer of commercially available drug libraries into a suitable screening format poses huge handling and logistics challenges. The parallel addition of reagents and solutions, and the transfer to another platform requires many multi-pipetting steps and thus large amounts of consumables as well as the loss of material due to large dead volumes. The handling of very large drug libraries is often impossible for a single individual to accomplish within a reasonable time. Additional labor costs make the development of bioactive compounds prohibitively expensive. Furthermore, commercially available drug libraries are extremely costly, and the number of available compounds remains limited. Synthesizing

new compounds is very time- and resource-consuming, since traditional solution-based organic synthesis is not designed for miniaturized and parallelized applications. In addition, the temporal and spatial separation of synthesis, characterization, and biological screening significantly slows down entire biological discovery and drug development pipelines. There is currently no system that could efficiently hyphenate these parts.

To solve these challenges, we developed a chemBIOS platform that unifies the miniaturized combinatorial solution-based organic synthesis in the microarray format and biological high-throughput screenings. The platform uses LSTL slides with omniphobic-omniphilic patterns to form arrays of organic solvents, where each droplet functions as a separate microreactor for solution-based organic synthesis. Furthermore, sandwiching the array with another droplet array enables the rapid and parallel addition, transfer, and copying of the whole library. For example, we carried out a three-component lipidoid combinatorial synthesis to create a library of 25 small molecules in triplicates (Fig. 3b). By sandwiching two droplet arrays, we simultaneously mixed all educts in 75 droplets and thus initiated the synthesis of an entire library in a single step (Fig. 3a). Although this platform can be readily utilized for many types of solution-based reactions using various organic or aqueous solvents, there are still some challenges in adapting other chemical techniques. Reactions at elevated temperatures and highly exothermic reactions as well as adding solid reagents or purging gaseous educts might be difficult

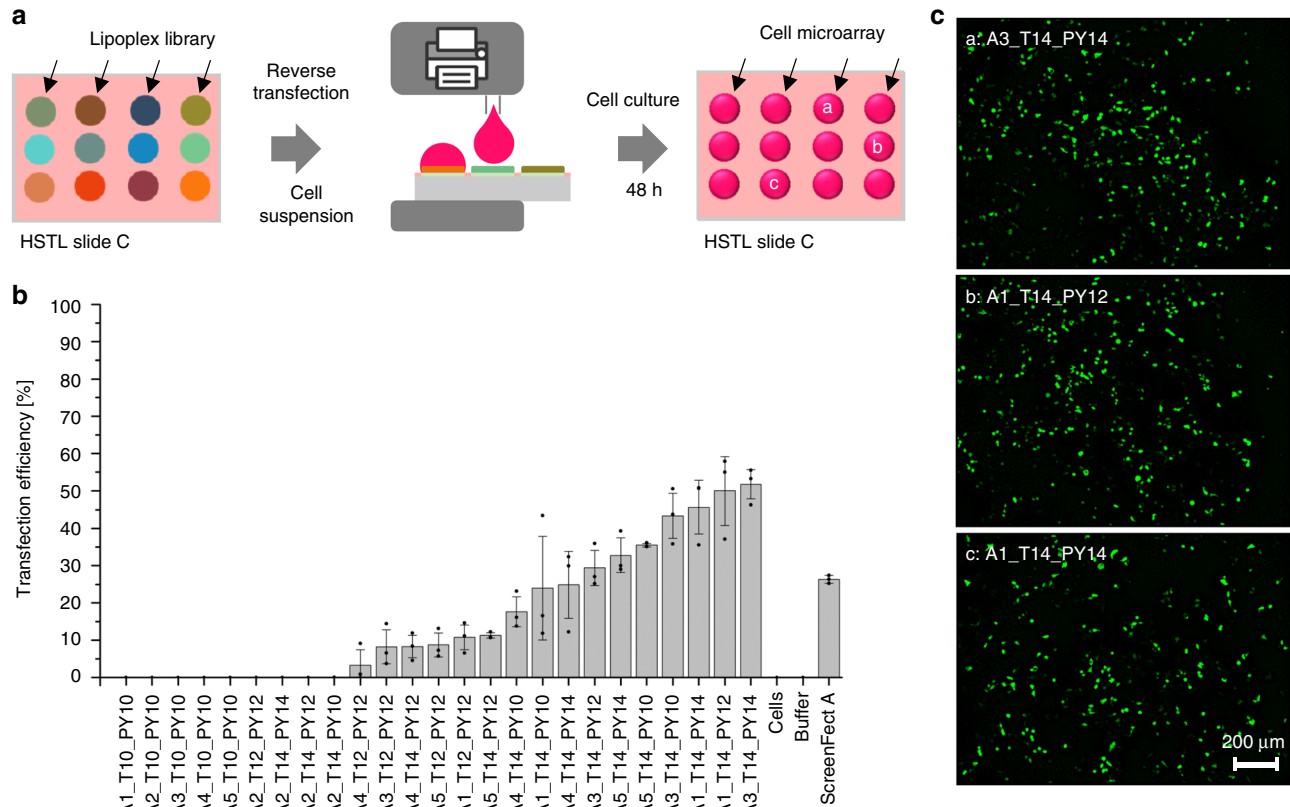

**Fig. 6** On-chip reversed cell transfection screening of the produced lipoplex library. **a** Schematic showing the process of on-chip cell transfection screening. Five microliter of HEK293T cell suspensions were printed into each hydrophilic spot on the HSTL slide C covered by the dried lipoplex cell transfection mixture. After incubating the array for 48 h at 36 °C and 5% $CO_2$, 1 μL of staining solution was printed into each droplet, followed by 15 min incubation and fluorescence microscopy analysis. **b** Bar chart visualizing mean values of transfection efficiencies (number of GFP-transfected cells per total cell number) calculated based on three independent experiments (overlaid dot plots) covering the entire pipeline including on-chip library synthesis, formation of lipoplexes and cell transfection. A3_T14_PY14 lipidoid proved to be most efficient in our study, resulting in 52 ± 4% transfection efficiency. Error bars are standard deviations, $n = 3$ (number of replicates); $N = 3$ (repetitions including lipid synthesis). Source data are provided as a Source Data file. **c** Fluorescence microscopy images of pCS2-GFP transfected cells. Scale bar: 200 μm

to realize using the chemBIOS platform in its current state. Chemical reactions requiring a protective atmosphere or controlled pressure are feasible, but would require a closed chamber with controlled pressure and atmosphere. On-chip parallel high-throughput purification is another key challenge that we will be focusing on in future studies. The advantages of the chemBIOS platform are that its configuration is open and flat, making all microcompartments accessible and potentially compatible with parallel high-throughput purification methods, e.g. on-chip extraction (Fig. 4a, b).

Due to the low sensitivity, most organic synthesis characterization methods are often incompatible with the micro-, nano- or even picomolar scale reactions required for high-throughput miniaturized synthetic applications. MALDI-TOF mass spectrometry is the method of choice for characterizing small quantities of compounds in a parallel way. The open system of the chemBIOS platform enabled us to copy the entire droplet-based library in a single step by stamping it onto a MALDI plate, followed by mass spectrometric analysis (Fig. 3c). Besides MALDI-TOF MS, compounds synthesized on the chemBIOS platform can be characterized via on-chip IR spectroscopy (Supplementary Fig. 8). This demonstrates another important advantage of the chemBIOS platform's flat, open droplet microarray system, which makes it compatible with other analytical methods such as Raman spectroscopy or

surface sensitive methods including DESI-MS or time-of-flight secondary ion mass-spectrometry.

We also demonstrated the feasibility of transferring the chemical compounds from the LSTL to a HSTL slide in a parallel manner that enables rapid and convenient hyphenation between the two realms—chemical and biological. We used HSTL slides to create highly hydrophilic-hydrophobic microarrays that could be used to form the arrays of aqueous droplets needed for biochemical and cellular screenings. We demonstrate the single-step transfer of the entire lipidoid library from the platform for organic solvents (LSTL slide) to the platform for aqueous solvents (HSTL slide) by simply sandwiching both types of slides. The single-step formation of liposomes and lipoplexes during this transfer step was proved by dynamic light scattering and zeta potential analysis—observing higher zeta potentials for liposomes than for lipoplexes (Fig. 5b). The lipoplexes we produced were successfully screened with cells by printing cellular suspension into the individual spots containing lipoplexes to perform a reversed cell transfection experiment (Fig. 6). The screening of lipidoids to determine their cell transfection efficiency using plasmid DNAs is an example of a biological assay that is combinable with synthesized libraries. Other possible biological experiments include, inter alia, CRISPR/Cas9-based gene editing, screening of membrane proteins, gene-knockdown, and the screening of cell spheroids or organoids. Overall, this

demonstrates the possibility to hyphenate combinatorial organic chemistry with biological screenings using the chemBIOS platform. Unifying chemistry and biology both temporally and spatially on the same platform accelerates the entire process of developing bioactive compounds important for various applications. Using the chemBIOS platform, in situ synthesized small molecule libraries can be screened immediately for their biological activity, thereby saving time, effort, chemicals, cells, and other consumables. The entire procedure of synthesizing (3 μL organic solvent), transferring/processing (4.5 μL aqueous solvent) and screening (5 μL cell suspension +1 μL staining solution) of a single lipidoid took only 3 days and about 13.5 μL of solutions, whereas the traditional bulk procedure requires at least several milliliters per compound. This allowed us to repeat the whole process from the synthesis to the biological screening multiple times; our final analysis of transfection efficiency by fluorescence microscopy yielded reproducible transfection results for all the library compounds (Fig. 6b). The platform is compatible with further miniaturization, despite possible future challenges associated with further reductions in droplet size, such as faster solvent evaporation, difficult alignment during the sandwiching process, compatibility with and sensitivity of analytical methods.

In summary, by unifying on-chip in situ miniaturized and parallelized solutions-based combinatorial synthesis of bioactive compounds, the feasibility of rapid and parallel characterization by copying the library microarray onto a separate slide for analysis (e.g. MALDI-MS), and on-chip biological and cellular screenings, the chemBIOS platform hyphenates all aspects of early-stage drug discovery, which will be useful for various other biological or biotechnological screening applications.

## Methods

**Preparation of high surface tension liquids slides**. Activation of the surface of standard microscope glass slides (25 × 75 × 1 mm, width × length × thickness, Schröder Spezialglas) was done by immersing them in 1 M NaOH (Carl Roth) for 1 h, followed by neutralization in 1 M HCl (Merck) for 30 min. Activated glass slides were modified with 20% v/v solution of 3-(trimethoxysilyl)propyl methacrylate (Sigma-Aldrich) in ethanol (Merck) for 40 min at room temperature. Next, 35 μL of polymerization mixture (24 wt% 2-hydroxyethyl methacrylate (Sigma), 16 wt% ethylene dimethacrylate (Sigma-Aldrich), 12 wt% 1-decanol (Merck), 48 wt% cyclohexanol (Sigma), and 0.4 wt% 2,2-dimethoxy-2-phenylacetophenone (Sigma-Aldrich)) was applied onto a fluorinated glass slide and covered with a modified glass slide to introduce a polymer layer. Polymerization was carried out by UV irradiation (OAI model 30) with 4 mW cm⁻² intensity and 260 nm wavelength for 15 min. Fluorination of glass slides was done by incubating slides in a closed vacuumed desiccator in the presence of an open vial containing 30 μL of trichloro(1H, 1H, 2H, 2H-perfluorooctyl)silane (Sigma) for 16 h under 50 mbar vacuum. Modification of the polymer layer was done by incubating the slides in modification mixture (56 mg of 4-(dimethylamino)pyridine (Novabiochem), 111.6 mg 4-pentynoic acid (Sigma-Aldrich) and 180 μL N,N'-diisopropylcarbodiimine (Alfa Aesar) in 45 mL dichloromethane (Merck)) for 4 h while stirring at room temperature. The surface was patterned by first generating the hydrophobic borders. Three hundred microliter of a 8% v/v solution of 1H, 1H, 2H, 2H-perfluorodecanethiol (Sigma-Aldrich) in acetone (Merck) was applied onto the polymer surface and the thiol-yne click reaction carried out by irradiating the slide through a photomask (Rose Fotomasken) with 260 nm UV light (OAI model 30) at 4 mW cm⁻² intensity for 1 min. Round hydrophilic spots with a diameter of 2.83 mm were formed by applying 200 μL of a 10% v/v β-mercaptoethanol (Alfa Aesar) solution in 1:1 water:ethanol onto the patterned surface and irradiating the slide with 260 nm, UV light (OAI model 30) at 4 mW cm⁻² intensity for 1 min.

**Preparation of low surface tension liquids slides**. An activated glass slide was modified by inserting into modification mixture (0.8 mL triethylamine (VWR), 50 mg of 4-dimethylaminopyridine (Merck), 0.2 mL chloro(dimethyl)vinyl silane (Sigma-Aldrich) in 49 mL dichloromethane) for 2 min under stirring at room temperature. The slide surface was patterned by first applying 300 μL of a 20% v/v solution of 1H, 1H, 2H, 2H-perfluorodecanethiol (Sigma-Aldrich) in acetone onto the modified surface and carried out a thiol-ene photoclick reaction by irradiating the slide through a photomask (Rose Fotomasken) with 260 nm UV light (OAI model 30) at 4 mW cm⁻² intensity for 1 min to create omniphobic borders. Round omniphilic spots with a diameter of 2.83 mm were formed by applying 200 μL of a 10 wt% cysteamine hydrochloride (Alfa Aesar) solution in 1:2 water:ethanol onto

the patterned surface and irradiating the slide with 260 nm, UV light (OAI model 30) at 4 mW cm⁻² intensity for 1 min.

**Surface characterization**. The surface of each slide was characterized by contact angle measurements using Drop Shape Analyzer DSA25 (Krüss). For HSTL slides we measured the advancing, static and receding contact angle by applying 40 μL (speed 0.3 μL s⁻¹) deionized water on hydrophobic and hydrophilic surfaces. For LSTL slides we measured the advancing, static and receding contact angle by applying 40 μL (speed 0.3 μL s⁻¹) DMSO (VWR) and deionized water on omniphobic and omniphilic surfaces.

**Synthesis and characterization of a lipidoid library**. On-chip synthetic procedure was performed in a standard chemistry fume hood. Stock solutions of different amines (Sigma-Aldrich) 1:24 v/v in DMSO and stock solutions of mixtures of different thiolactones (1.67 mg mL⁻¹) and pyridyl disulfides (1.75 mg mL⁻¹) in DMSO were prepared. The on-chip synthesis of a lipidoid library via the rolling droplet method was performed by applying 3 μL of different amine solutions to each spot column-by-column in an array on a LSTL slide A and 3 μL of different mixtures of thiolactones and pyridyl disulfides to each spot row-by-row in an array on a LSTL slide B perpendicular to the columns on slide A by discontinuous dewetting. The reaction was carried out by sandwiching both slides using an alignment frame for 2 h at room temperature. For characterization, the synthesized library was stamped onto a MALDI plate (Applied Biosystems). Solvent was evaporated in a desiccator under vacuum. Mass spectrum of each raw product was measured by 4800 MALDI TOF/TOF Analyzer (Applied Biosystems) using a 10 mg mL⁻¹ α-cyano-4-hydroxycinnamic acid (Alfa Aesar) solution of 1:1 acetonitrile:water containing 0.1% v/v trifluoroacetic acid (Merck). Further characterization by UV-Vis was done by Lambda 35 UV-Vis spectrometer (PerkinElmer). On-chip ATR-IR spectroscopy was done by Tensor 27 (Bruker). For characterization by ¹H-NMR spectroscopy (Bruker 400 MHz), we synthesized one lipidoid on a whole array as described above, then evaporated the solvent and washed down the raw product with dichloromethane (Merck) from LSTL slides into a round bottle flask. We evaporated the solvent under high vacuum and dissolved the raw product in deuterated chloroform (VWR) for analysis.

The on-chip synthesis of a lipidoid library via the printing method was performed by printing 1.5 μL of different amine solutions to each spot column-by-column in an array on a LSTL slide A using a non-contact liquid dispenser (I-DOT; Dispendix). Next, 1.5 μL of different mixtures of thiolactones and pyridyl disulfides were printed to each spot row-by-row on the same array perpendicular to the columns of previously printed amine solutions using a non-contact liquid dispenser (I-DOT; Dispendix). The reaction was carried out for 2 h at room temperature before the solvent was evaporated in a desiccator under vacuum.

This synthesizing method was used for all experiments except the characterization experiments by MALDI TOF MS.

**On-chip purification**. On-chip purification was performed in a standard chemistry fume hood: Three microliter per spot of a mixture of Nile red (2.1 mg mL⁻¹; Sigma-Aldrich) and methylene blue hydrate (2.7 mg mL⁻¹; Thermo Fisher) in 1-octanol (Sigma-Aldrich) were applied to several spots of an LSTL slide. Five microliter per spot of deionized water was applied to several spots on a HSTL slide corresponding to the spots on the LSTL slide. Both slides were sandwiched for 10 min and then separated. One microliter of both the organic and aqueous phase of each spot were diluted separately in 1 mL acetonitrile for characterization by a Lambda 35 UV-Vis spectrometer (PerkinElmer). One microliter of both the organic and aqueous phase of each spot were diluted separately in 50 μL acetonitrile for characterization by an Agilent 1100 LC/MS (Agilent Technologies).

**Preparation and characterization of liposomes/lipoplexes**. On-chip preparation of liposomes/lipoplexes was performed under sterile conditions using a standard sterile clean bench: A sterile filtered (0.45 μm sterile syringe filter) transfection solution of 0.04% w/v gelatin (Sigma-Aldrich), 3.4% w/v sucrose (Sigma-Aldrich), 0.002% w/v human fibronectin (Sigma-Aldrich) and—in case of formation of lipoplexes—plasmid DNA (75 ng μL⁻¹ of pCS2+ GFP) in aqueous sodium acetate buffer (50 mM, pH 5, Merck) was prepared for post-synthetic processing of the lipidoid library to form a liposome or lipoplex library. In all, 4.5 μL of prepared transfection mixture was printed onto each spot of a HSTL slide C by a non-contact liquid dispenser (I-DOT; Dispendix), followed by sandwiching the slide C with dried LSTL lipidoid library slide A for 1.5 h at 50 °C using an alignment frame to form liposomes or lipoplexes. For characterization via dynamic light scattering (DLS) and zeta potential, 2 μL of processed liposome/lipoplex solution of a single spot was pipetted into 1 mL sodium acetate buffer (50 mM, pH 5) for analysis by Malvern Zetasizer Nano ZS (Malvern). Data were collected at 25 °C with an acquisition time of 15 s and the diameter size was averaged over 3 × 15 runs.

For positive control we used ScreenFect dilution buffer containing 0.04% w/v gelatin, 3.4% w/v sucrose and 0.002% w/v human fibronectin. In a PCR tube we mixed 12.6 μL of ScreenFect A (Screenfect) into 23.7 μL ScreenFect dilution buffer and incubated for 5 min at room temperature. In a separate PCR tube, we mixed

1.67 µL plasmid DNA (2.05 µg µL⁻¹ of pCS2 + GFP) into 6.93 µL ScreenFect dilution buffer. The ScreenFect A solution was mixed into the plasmid DNA and incubated for 20 min at room temperature. Thus, led into to formation of lipoplexes with the same calculated pDNA to lipid ratio of our synthesized samples of 0.273 µg µL⁻¹. We applied 4.5 µL of prepared ScreenFect solution to unused spots of the library array after processing of the lipidoid library to liposome/lipoplex library. For negative control we applied 4.5 µL buffer solution to unused spots after liposome/lipoplex formation. Furthermore, we also screened spots without any compounds for negative control. All spots were dried under atmosphere pressure and sterile conditions for 2 h before continuing with cell seeding for reverse transfection.

**Reverse transfection and cell culture**. On-chip reverse transfection was performed under sterile conditions using a standard sterile clean bench: For our transfection experiments, we used human embryonic kidney (293T (ATCC® CRL3216™)) cells provided by the Institute of Toxicology and Genetics (ITG) at Karlsruhe Institute of Technology (KIT). Cells were cultured in Dulbecco's Modified Eagle Medium (Life Technologies) supplemented with 10% v/v fetal bovine serum (PAA Laboratories) and 1% v/v penicillin-streptomycin (Life Technologies) in a humid incubator at 37 °C with 5% CO₂, and were passaged every 2–3 days. Cells were washed with phosphate buffered saline (Life Technologies), detached with 0.25% trypsin/EDTA solution (Life Technologies) and counted by Countess Automated Cell Counter (Life Technologies). Cells were stained by mixing 10 µL cell suspension into 10 µL trypan blue stain solution (Life Technologies). Ten microliter of this mixture were applied into cell counting chamber (Life Technologies) for counting the cells.

Culture medium for on-chip transfection experiments contained 15% v/v fetal bovine serum and 1% v/v PenStrep. After detachment of the cells from culture plate, the cells were centrifuged at 1200×g for 3 min at room temperature to form a cell bead. The old medium was removed and then the cells resuspended again with freshly prepared medium. Cell suspension containing 60,000 cells mL⁻¹ were prepared and 5 µL of that suspension was printed onto each spot of the lipoplex slide C using a non-contact liquid dispenser (I-DOT; Dispendix). We placed the seeded cell slide in a Petri dish whose lid we had prepared with a 7 mL phosphate buffer saline soaked sterile tissue (Clean and Clever)—to prevent media evaporation of the droplets. The whole Petri dish was then placed in a humid incubator for 48 h at 37 °C with 5% CO₂. After culturing the cells for reverse transfection, cells were stained by dispensing 1 µL staining solution onto each spot of the array containing Hoechst 33342 in dilution of 1:900 (10 mg mL⁻¹, Invitrogen) to stain the nucleus and propidium iodide in a dilution of 1:1350 (1.00 mg mL⁻¹, (Invitrogen)) to stain dead cells.

**Image acquisition and analysis**. Fluorescence images were obtained using the microscope Keyence BZ9000 (Keyence). The exposure times were set and kept the same for all experiments and repetitions. Images were taken using a ×10 objective.
Objective: Nikon ×10 Plan Apo NA 0.45/4.00 mm
Resolution: 8-bit
Format: 1360 × 1024 px
Light source: mercury vapour lamp
Cells were counted by adjusting the threshold of the 8-bit images and then using the Analyze Particles function in ImageJ. The ration of transfected cells in GFP channel and total number of cells in Hoechst channel revealed transfection efficiency of a sample. All date sets were depicted as mean ± standard deviation. At least three replicates per slide and at least three slides were tested for each transfection sample.

**Reporting summary**. Further information on research design is available in the Nature Research Reporting Summary linked to this article.

## Data availability
The data that support the findings of this study are available from the corresponding author upon reasonable request. The source data underlying Figs. 3c, 4b, and 5b and Supplementary Figs. 1, 5a–c, 6a–d, 7, 8, 9, 10a–e, 11a–b, and 12 and Supplementary Tables 1, 2 and 3 are provided as a Source Data file.

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

## Acknowledgements

The research was supported by the ERC Starting Grant (DropCellArray, 337077), ERC Proof-of-Concept Grant (CellScreenChip, 680913) and the Ministry of Science, Research and the Arts Baden-Württemberg (MWK), Grant FKZ 7533-7-11-10-07 (IWWB-28). We would like to thank Dr. Gerald Brenner-Weiß from IFG (KIT, Institute of Functional Interfaces) for providing training and access to MALDI-TOF mass spectrometry, and Dr. Nicole Jung from IOC (KIT; Institute of Organic Chemistry) for providing training and access to LC-MS.

## Author contributions

P.A.L. proposed the initial idea and elaborated it together with M.B.; M.B. designed and constructed the devices, designed the experimental setup, performed the experiments, analyzed, and interpreted the data; M.R.M. synthesized the precursor molecules; M.B. and A.B. performed MS characterization; M.B., A.R., and P.A.L. wrote the manuscript with input from all authors.

## Additional information

**Competing interests:** The authors declare no competing interests.

