## [Peer Review File · Nature Communications]

Reviewers' comments:

Reviewer #1 (Remarks to the Author):

Drug lead synthesis and screening is an arduous and time-intensive endeavor that is made more challenging by the low success rate in translating a lead into clinical testing and commercialization. Accessing chemical space rapidly in high-throughput screening is a strategy to overcome these limitations, which benefits from miniaturization and automation. Here, Benz, et al. describe a method for rapid on-chip, solution-phase rapid parallel organic synthesis of a lipidoid library, MALDI-MS characterization, on-chip liposome formation, and on-chip cellular screening of transfection efficiency of said liposomes. The authors have designed an on-chip platform that allows for the entire process was completed rapidly, and requires approximately 1 mL of total solvents. This manuscript has many strengths that include: 1) a solution-phase small molecule microarray-like system that represents an advance from previous methods in that "compounds" (here lipidoids) can be screened in solution; 2) a system to modulate droplet size on a microarray slide; 3) a method to quickly characterize the compounds by transferring the droplet to a MALDI-MS plate; and 4) (though not discussed), a potential system to screen membrane proteins in the droplets produced, which would represent a huge advance in the field. This latter point is not proven, but demonstration of this capability would improve the impact of the manuscript significantly.

Unfortunately, the authors claim, yet fail to demonstrate, that these methods are applicable to both synthesis and screening of a broad range of ligands and targets. By focusing on screening lipidoids for transfection efficiency, this technology is likely only applicable to a small group of researchers. In addition, the chemical transformations do not appear to be robust (see below). Finally, there are other scientific concerns that limit enthusiasm for publication in Nature Communications. Therefore, unless more broad applications of this technology can be demonstrated in a new manuscript, and the concerns below are met, this reviewer does not feel that the merit of this manuscript rises to the level of the journal and may be more appropriate for a more specialized journal.

Some of the major concerns:

1. While design of the LSTL and HSTL chips features an elegant design and the supplementary video is impressive, a discussion of the scope of chemical reactions that can be performed is lacking. In order for this approach to have true potential as a drug discovery platform as claimed, the authors should demonstrate other chemical transformations that are possible on the chemBOIS chip and address other compatible solvents (not just DMSO). It should also be mentioned that this method may suffer from inconclusive screening results as there are no purification steps. Therefore, it is presumed by the authors that high fidelity reactions are a requirement. While the 2-thiopyridone and other impurities produced in this reaction may not have impacted the GFP transfection assay, that may not be true for the synthesis of other scaffolds, screening of other targets, or other phenotypic assays.
2. This platform's open droplet microarray displays clear advantages in chip-to-chip transfer. Single step transfer to MALDI is important for rapid characterization, however it should be stated that MALDI may not provide an accurate assessment of reaction yield due to variation in ionization and that NMR and/or liquid chromatography should be performed on hit compounds.
3. The spectroscopic approach taken to monitor reaction progress in this study may lack translatability to other chemical reactions.
4. Experiments concerning liposome preparation and GFP transfection assay were well done, but as mentioned above, this is a limited application.
5. The introduction is written like a highly selective review of drug discovery. However, there are few chosen examples that are described to make the authors' point drug screening is tedious and time consuming. This list is not comprehensive, and probably inappropriate here. Much of this can be removed or revised.

Reviewer #2 (Remarks to the Author):

This manuscript describes the on-chip platform for synthesis, characterization and biological screening demonstrating the potential for the drug development. The author synthesized and screening lipidoids as transfection agent to proof the concept. Taken together, I recommend publication of a small number of concerns have been addressed:

(1) As shown in Figure 4, the zeta-potential of some lipoplexes appeared to be negative charge. How does the author concern about the N/P value for the lipoplexes formation?

Reviewer #3 (Remarks to the Author):

Levkin and coworkers have developed an interesting technology for the rapid synthesis of chemical libraries that can be screened structurally and, in some cases, biologically. By modifying the chemistry of a slide surface, it is possible to control the introduction of specific reagents, both chemical and biological. Overall, this is an interesting study, although there are some concerns that need to be addressed.

1. Figure 1 should only be the right-most figure, as the other parts are already well known.
2. The authors indicate that there is a "methodological gap" between synthesis and screening and that has been a major cause of a lack of progress in drug development efficiency. I do not agree. The problem has typically been downstream – poor lead optimization, limited specificity, potential toxicity, etc.
3. There have been myriad combichem approaches developed over the past 20+ years. Rapid screening was performed, even biological screening. The argument over chemical incompatibility is weak.
4. The lack of academic studies is not so much due to the cost as it is the goal within a non-commercial environment. Fundamental studies do not typically require very large libraries. Moreover, combinatorial libraries are losing favor to much more focused libraries that can interrogate a much broader structurally universe.
5. I like the synthetic scheme of Slide 3 that allows rapid synthesis, minimal cross-talk, and then off-line analysis on the chip (e.g., MALDI). But, off-line analysis still required if there is a need to know if mixtures exist and the approximate conversions.
6. A technical issue should be addressed. In the schemes, two chips are contacted through liquid-liquid contact. For this to be successful, there needs to be very good alignment of the two chips. Anything off even several microns (depends on spot size) will reduce effectiveness. How is this accounted for by the authors?
7. The ability to include biological reagents (such as plasmid DNA) is quite useful. The authors showed improved transfection, albeit with higher toxicity. This opens up opportunities for downstream applications, e.g., CRISPR. That may need to be addressed in more detail.

Reviewer #4 (Remarks to the Author):

The ms describes an essentially novel microarray droplet technology based on patterns with different types of wettabilities and chemical or biological compounds/cells allowing for the synthesis of model chemical compounds, converting them into an aqueous droplet array and finally testing a library of

compounds in a cellular assay (here transfection).

The technology builds partly on microarray drop array techniques developed earlier in the KIT group, but the combination is entirely new in my view, addressing the highly important fabrication of chemical/pharmaceutical libraries and its evaluation in cell-based assays. This microarray microdroplet technique is highly innovative and has the potential of massively shortening the time and reducing the amount of compounds needed for library synthesis, microarray preparation, and cell-based highly parallel readouts.

The presentation of the result is very professional including state of the art, clear argumentation, background info, process development and readouts, nicely designed figures, and references.

A topic that is (I believe) not addressed is the issue of byproducts produced during the library synthesis of chemical compounds. Although turnover is quite impressive, small amounts of side products might have important undesirable effects such as cell toxicity. This should be addressed. Also, the issue of what clean room standards would have to be used in a commercial fab lab would be useful to know.

Otherwise, I highly recommend publication after the proposed revisions. I believe that this publication will be of great interest to a large community in both academia and industry.

Response to reviewer's comment

Reviewer #1:

Comment: Drug lead synthesis and screening is an arduous and time-intensive endeavor that is made more challenging by the low success rate in translating a lead into clinical testing and commercialization. Accessing chemical space rapidly in high-throughput screening is a strategy to overcome these limitations, which benefits from miniaturization and automation. Here, Benz, et al. describe a method for rapid on-chip, solution-phase rapid parallel organic synthesis of a lipidoid library, MALDI-MS characterization, on-chip liposome formation, and on-chip cellular screening of transfection efficiency of said liposomes. The authors have designed an on-chip platform that allows for the entire process was completed rapidly, and requires approximately 1 mL of total solvents. This manuscript has many strengths that include: 1) a solution-phase small molecule microarray-like system that represents an advance from previous methods in that “*compounds*” (here lipidoids) can be screened in solution; 2) a system to modulate droplet size on a microarray slide; 3) a method to quickly characterize the compounds by transferring the droplet to a MALDI-MS plate; and 4) (though not discussed), a potential system to screen membrane proteins in the droplets produced, which would represent a huge advance in the field. This latter point is not proven, but demonstration of this capability would improve the impact of the manuscript significantly.

Unfortunately, the authors claim, yet fail to demonstrate, that these methods are applicable to both synthesis and screening of a broad range of ligands and targets. By focusing on screening lipidoids for transfection efficiency, this technology is likely only applicable to a small group of researchers. In addition, the chemical transformations do not appear to be robust (see below). Finally, there are other scientific concerns that limit enthusiasm for publication in Nature Communications. Therefore, unless more broad applications of this technology can be demonstrated in a new manuscript, and the concerns below are met, this reviewer does not feel that the merit of this manuscript rises to the level of the journal and may be more appropriate for a more specialized journal.

While design of the LSTL and HSTL chips features an elegant design and the supplementary video is impressive, a discussion of the scope of chemical reactions that can preformed is lacking. In order for this approach to have true potential as a drug discovery platform as claimed, the authors should demonstrate other chemical transformations that are possible on the chemBOIS chip and address other compatible solvents (not just DMSO).

Response: We thank the reviewer for highlighting the strength of our work. The point raised by the reviewer regarding the screening of membrane protein is very interesting. We plan to do the experiment in the extension work of this project. Here we describe a method to miniaturize and parallelize solution-based combinatorial synthesis and demonstrate how to combine it with biological screening. As a proof of concept, we performed only one type of reaction and we plan to do many more in future work.

In our model reaction (synthesis of lipidoids) we indeed utilized just one organic solvent (DMSO). As correctly stated by the reviewer, other chemical reactions might need other solvents. In order to demonstrate the applicability of the chemBIOS platform to other types of solvents and, thus, chemical transformations, we performed an additional experiment to test the possibility to form droplets confined to the omniphilic spots using various commonly used organic solvents such as

n-hexane ($\gamma_{lv} = 18.4 \text{ mN m}^{-1}$), ethanol ($\gamma_{lv} = 22.1 \text{ mN m}^{-1}$), 1-decanol ($\gamma_{lv} = 28.5 \text{ mN m}^{-1}$) and DMF ($\gamma_{lv} = 37.1 \text{ mN m}^{-1}$):

Supplementary Figure 2 | Photograph of droplets of various organic solvents on a Low Surface Tension Liquids (LSTL) slide. From left to right: n-hexane ($\gamma_{lv} = 18.4 \text{ mN m}^{-1}$), ethanol ($\gamma_{lv} = 22.1 \text{ mN m}^{-1}$), 1-decanol ($\gamma_{lv} = 28.5 \text{ mN m}^{-1}$), DMF ($\gamma_{lv} = 37.1 \text{ mN m}^{-1}$) and DMSO ($\gamma_{lv} = 43.5 \text{ mN m}^{-1}$). Spot size: 2.83 mm; borders width: 1.67 mm; droplet volume: 5 μL .

Thus, the platform can be readily utilized for different types of solution-based reactions using various organic solvents. Nevertheless, we are aware of the challenges to adapt some other typical procedures utilized in organic synthesis. For example, reactions at elevated temperature, utilizing solid reagents, highly exothermic reactions will be still challenging using the chemBIOS platform. In our future and ongoing research, we will evaluate the possibility to perform various types of reactions, as well as various types of purification procedures common for organic chemistry (e.g. high-throughput crystallization, extraction and even sublimation).

We added the following text to the discussion part:

“Despite the fact, that the platform can be readily utilized for many types of solution-based reactions using various organic or aqueous solvents, there are still some challenges in adapting other chemical techniques. Reactions at elevated temperatures and highly exothermic reactions as well as adding solid reagents or purging gaseous educts might be difficult to realize using the chemBIOS platform in its current state. Chemical reactions requiring a protective atmosphere or controlled pressure can be performed but would require a closed chamber with controlled pressure and atmosphere.”

Comment: It should also be mentioned that this method may suffer from inconclusive screening results as there are no purification steps. Therefore, it is presumed by the authors that high fidelity reactions are a requirement. While the 2-thiopyridone and other impurities produced in this reaction may not have impacted the GFP transfection assay, that may not be true for the synthesis of other scaffolds, screening of other targets, or other phenotypic assays.

Response: We agree with the reviewer that byproducts might have an impact on the biological assays used in the screening and, although we did not see such an effect in our transfection screening, on-chip purification will be essential for the future development of the technology. We are currently working on this issue and the advantage of the open system of the droplet microarrays is that it is compatible with various purification methods commonly used in solution based organic chemistry, such as high-throughput parallel crystallization, extraction, sublimation and even centrifugation. In order to demonstrate the feasibility of such high-throughput parallel purification, we performed an additional experiment showing a proof-of-principle on-chip two-phase extraction. We added the following part to the result part of the manuscript:

“The open format of the chemBIOS system is compatible with further on-chip steps, such as investigation, treatment or conversion of the chemical compound library, to achieve high-throughput parallel on-chip purification. In a proof-of-principle experiments, we showed how crude products could be purified by an on-chip two-phase liquid extraction (**Fig. 4a,b**). Therefore, we dissolved Nile red and methylene blue in 1-octanol which resulted in a dark blue solution. Next, we applied this solution on several spots on an LSTL slide D and sandwiched this slide with an HSTL slide E containing water droplets. Sandwiching of both solvents resulted in the formation of an interface between the solvents since they are not miscible (**Fig. 4a**). Methylene blue is highly soluble in water and therefore was extracted from the organic phase into the aqueous phase. Nile red is insoluble in water and, therefore, remained in the organic phase. After 10 min the color of organic phase turned red, while the color of aqueous phase turned blue, indicating a successful separation of both dye compounds.

Figure 4 | On-chip parallel liquid-liquid extraction. (a) Schematic describing of the process of on-chip two-phase liquid extraction. A mixture of oleophilic Nile red and hydrophilic methylene blue in 1-octanol on an LSTL slide D was separated by sandwiching the slide with an HSTL slide E carrying droplets of water for 10 min. An interface between both non-miscible solvents was formed and the water-soluble methylene blue was extracted into the aqueous phase. (b) Validation of the on-chip extraction by UV-Vis spectroscopy. The UV-Vis spectrum of the mixture shows two local absorbance maxima, one at 541 nm which corresponds to Nile red and another at 654 nm corresponding to methylene blue. After the extraction, the intensity of the absorbance maximum of methylene blue in the organic phase decreased while the intensity of Nile red remained constant. In the aqueous phase, no Nile red could be detected. Source data are provided as a Source Data file.

We analyzed the mixture before purification, as well as both the organic and aqueous phase after the purification by UV-Vis spectroscopy (**Fig. 4b**) and liquid chromatography mass spectrometry (LC-MS) (**Supplementary Fig. 9 and 10**) to estimate the success of purification. Both analysis methods showed comparable results. While Nile red remains dissolved only in the organic phase, a significant amount of methylene blue was extracted into the aqueous phase. Analysis by UV-Vis spectroscopy enables quantification of this result by the use of Beer-Lambert law. Therefore, the extinction coefficients of Nile red and methylene blue in acetonitrile were calculated to be $43,506 \text{ M}^{-1} \text{ cm}^{-1}$ at 541 nm and $72,653 \text{ M}^{-1} \text{ cm}^{-1}$ at 654 nm, respectively (**Supplementary**

Fig. 11). We estimated the concentration of methylene blue to be 11.9 ± 0.5 mM in the original organic phase mixture, while 3.3 ± 0.6 mM and 4.1 ± 0.4 mM in the organic and aqueous phase, respectively, after extraction. The concentration of Nile red was estimated to be 6.0 mM in the original mixture, while 4.3 ± 0.3 mM and 0.3 ± 0.04 mM in the organic and aqueous phase after the extraction. This corresponds to the purification of the methylene blue from 66% in the mixture to 91% in the aqueous phase and that of Nile red from 34% in the mixture to 57% in the organic phase by a single purification step.”

Furthermore, we complemented the discussion part by adding the following:

“The on-chip parallel high-throughput purification is another very important challenge on that we have to focus in further studies. The advantages of the chemBIOS platform are that it possesses open and flat configuration making all microcompartments accessible and potentially compatible with parallel high-throughput purification methods, e.g. on-chip liquid-liquid extraction (**Fig. 4a,b**).”

The peculiarity of the lipidoid reaction we presented here is that it produces the same byproduct for all the lipids. Once the best lipids are identified, they can be synthesized in the bulk, purified accordingly and then can be used for more focused secondary screenings, where purity of the product is an important issue. The aim of a primary screening – which was demonstrated by our chemBIOS system – is to accelerate and facilitate the primary synthesis and screening of large compound libraries in short time.

Comment: *This platform's open droplet microarray displays clear advantages in chip-to-chip transfer. Single step transfer to MALDI is important for rapid characterization, however it should be stated that MALDI may not provide an accurate assessment of reaction yield due to variation in ionization and that NMR and/or liquid chromatography should be performed on hit compounds.*

Response: We agree with the reviewer that MALDI-MS is not always quantitative due to the difference in ionization. However, we did not use the MALDI-MS to assess reaction yield. We used UV-Vis spectroscopy to estimate the reaction yield by measuring the absorption of the by-product 2-thiopyridone. The presence of the correct compounds was confirmed by MALDI-MS characterization. Due to the fact that the platform is open, it is also possible to use other

characterization methods, such as LC-MS. We performed another experiment showing the possibility to separate and characterize compounds by LC-MS.

We added the following figures to the supplementary information:

Supplementary Figure 9 | HPLC-MS characterization of the on-chip liquid-liquid extraction process. A mixture of a solution of Nile red and methylene blue was purified by an on-chip two-phase liquid extraction. The mixture (black line), the organic (red line) and the aqueous phase (blue line) was analyzed by HPLC-MS. Methylene blue showed a retention time of about 5.7 min, Nile red showed a retention time of 11.1 min. The intensity of methylene blue in the organic phase decreased after the purification step, while the intensity of Nile red remained the same. No peak for Nile red could be detected in the aqueous phase. Flow rate: 1 mL min⁻¹; detector: DAD (230 nm, 254 nm, 280 nm, 300 nm and 400 nm). Source data are provided as a Source Data file.

Supplementary Figure 10 | LC-MS spectra. (a) Mixture; retention time: 5.7 min – calculated mass (m/z): 284.12; found mass (m/z): 284.10. **(b)** Mixture; retention time: 11.1 min – calculated mass (m/z): 318.14; found mass (m/z): 319.10. **(c)** Organic phase; retention time: 5.8 min – calculated mass (m/z): 284.12; found mass (m/z): 284.10 **(d)** Organic phase; retention time: 11.1 min – calculated mass (m/z): 318.14; found mass (m/z): 319.10. **(e)** Aqueous phase; retention time: 5.7 min – calculated mass (m/z): 284.12; found mass (m/z): 284.10. Source data are provided as a Source Data file.

Furthermore, we complemented another on-chip characterization method by on-chip ATR-IR spectroscopy and added the following figure to the supplementary information, as well:

Supplementary Figure 8 | Exemplary IR spectrum of the product A5_T14_PY14. Source data are provided as a Source Data file.

Comment: The spectroscopic approach taken to monitor reaction progress in this study may lack translatability to other chemical reactions.

Response: We agree with the reviewer that there are many analytical methods (UV-Vis, IR and Raman spectroscopy, LC-MS, TLC etc.) one can follow to monitor an organic reaction. Here, we chose the most appropriate approach for this particular reaction, which is UV-Vis technique as the by-product is UV active. There is no restriction for this platform in choosing analytical method to monitor the progress of a reaction. One needs to choose the method depending on the nature of the reactions. In addition, there are various surface-sensitive mass-spectrometric and other methods that are compatible with the chemBIOS platform: e.g. LESA-MS, DESI-MS, ToF-SIMS, XPS, etc.

We added the following part to the discussion as a combination of this reviewer's comment and the previous one:

“The open and flat microarray format of the chemBIOS slides makes it compatible with further analytical methods such as IR or Raman spectroscopy or surface sensitive methods including DESI-MS or time-of-flight secondary ion mass-spectrometry.”

Comment: Experiments concerning liposome preparation and GFP transfection assay were well done, but as mentioned above, this is a limited application.

Response: Here, we showed a model multicomponent reaction, liposome preparation and screening for protein expression using droplet microarrays. This is not the only application possible using this platform. As we have shown, different types of organic solvents can form droplets on this surface, indicating that it is possible to do various reactions on this platform. In our group, we do various types of multicomponent synthesis on DMA and combine them with biological assays using different types of cells. We also added a discussion in the manuscript about other potential application such as screening of membrane proteins.

Comment: The introduction is written like a highly selective review of drug discovery. However, *there are few chosen examples that are described to make the authors' point drug screening is tedious and time consuming.* This list is not comprehensive, and probably inappropriate here. Much of this can be removed or revised.

Response: The introduction part has been revised according to the reviewer's comment.

Reviewer #2:

Comment: This manuscript describes the on-chip platform for synthesis, characterization and biological screening demonstrating the potential for the drug development. The author synthesized and screening lipidoids as transfection agent to proof the concept. Taken together, I recommend publication of a small number of concerns have been addressed:

Response: We thank the reviewer for recommending publication of our work.

Comment: As shown in Figure 4, the zeta-potential of some lipoplexes appeared to be negative charge. Have the author concern about the N/P value for the lipoplexes formation?

Response: We agree with the reviewer that lipoplexes with zeta-potential close to zero or even negative are often considered suboptimal for cell transfection. On the other hand, very positive zeta-potential is considered suboptimal for in-vivo applications and it is important to search for efficient but neutral liposomes. We have calculated the N/P ratio and reported it in the supplementary information.

The following table was added to the supplementary information:

Supplementary Table 4 | Calculated N/P values of selected lipoplexes. Source data are provided as a Source Data file.

Sample	N/P value
A1_T10_PY10	5:1
A3_T12_PY12	8:1
A4_T14_PY12	4:1

Reviewer #3:

Comment: Levkin and coworkers have developed an interesting technology for the rapid synthesis of chemical libraries that can be screened structurally and, in some cases, biologically. By modifying the chemistry of a slide surface, it is possible to control the introduction of specific reagents, both chemical and biological. Overall, this is an interesting study, although there are some concerns that need to be addressed.

Response: We thank the reviewer for the positive response.

Comment: Figure 1 should only be the right-most figure, as the other parts are already well known.

Response: We thank the reviewer for this comment, but we think that we should keep both parts of the figure for a better visualization of the process of drug discovery and easier comparison to the described process.

Comment: The authors indicate that there *is a "methodological gap" between synthesis and screening* and that has been a major cause of a lack of progress in drug development efficiency. I do not agree. The problem has typically been downstream – poor lead optimization, limited specificity, potential toxicity, etc.

Response: We agree with the reviewer, that the lack of progress in drug development efficiency is also due to poor lead optimization, limited specificity, potential toxicity, etc. but these are not the only reasons for this issue. The problems with lead optimization, limited specificity, toxicity, mentioned by the reviewer, are to some extent the result of limited primary screenings. Due to the costs and availability, primary screenings are usually only done once (without repetitions and different concentrations) and only in big pharmaceutical companies or screening centers. The broader accessibility to such screenings (democratization) will inevitably lead to hits of "higher quality", i.e. better specificity, lower toxicity, etc. On the other hand, the downstream processes, such as time-consuming lead optimization, can also benefit from the chemBIOS platform directly. We have taken this into account and revised the introduction.

Comment: There have been myriad combichem approaches developed over the past 20+ years. Rapid screening was performed, even biological screening. The argument over chemical incompatibility is weak.

Response: We agree with the reviewer that a lot of research was performed in this field. However, what we wrote in our manuscript is that most of the organic chemistry methods developed so far, including many combichem methods, are incompatible with the biological screenings that are usually performed in polystyrene or COC microtiter plates (see pages 2 and 3). Most of the existing combi-hem methods designed for biological assays (e.g. solid-phase peptide or oligosaccharide arrays, DNA-encoded libraries, one-bead-one-compound, etc.) are not compatible with more physiologically relevant cellular assays (2D cell culture, 3D spheroid or even in vivo screenings) where freely diffusing compounds in an array format are needed. These methods require additional steps for releasing the compounds into the cell suspension whereby most releasing methods are not bioorthogonal (e.g. cleavage of the compounds under strong acidic or strong alkaline conditions or requiring other cell toxic treatments). Furthermore, the above-mentioned methods suffer from the lack of compartmentalization. These are the main challenges that we solve with our chemBIOS platform based on compartmentalized microdroplet reactions. Each droplet acts as a separated reaction space, where we can do miniaturized and parallelized solution-based synthesis. We revised the introduction part to make it clearer regarding this point:

“Moreover, most of the existing SPS methods are not compatible with more physiologically relevant cellular assays (2D or 3D cell culture) where freely diffusing compounds, and, therefore, compartmentalization of individual cell experiments, are required.”

Comment: The lack of academic studies is not so much due to the cost as it is the goal within a non-commercial environment. Fundamental studies do not typically require very large libraries. Moreover, combinatorial libraries are losing favor to much more focused libraries that can interrogate a much broader structurally universe.

Response: The method is in general applicable for both random combinatorial libraries, as well as for more focused libraries. We disagree that fundamental studies do not typically require very large libraries. The main reason why academic labs do not perform such screenings is the unaffordable costs of such libraries. The goal of this project is to develop a platform that is potentially able to make large compound screenings affordable to every academic lab in the future. Such “democratization” of screenings can have long standing important positive consequences on drug discovery and related fields.

Comment: I like the synthetic scheme of Slide 3 that allows rapid synthesis, minimal cross-talk, and then off-line analysis on the chip (e.g., MALDI). But, off-line analysis still required if there is a need to know if mixtures exist and the approximate conversions.

Response: We thank the reviewer for this comment. Off-line analysis is required at the reaction optimization stage and, of course, would be useful at the step of combinatorial screening of large libraries. However, the ultimate goal of this technology will be the synthesis of thousands or hundreds of thousands compounds in parallel (one microtiter plate-sized glass slide can accommodate several thousand droplets at the moment). At such a scale, it will be either impossible to analyze every compound by an off-chip HPLC or NMR not only due to the time constricts ($10.000 \text{ compounds} * 5 \text{ min analysis time} = 35 \text{ days}$) but also due to the sensitivity

limitations. In this respect, the method can be compared with peptide microarrays or DNA microarrays where only on-chip high-throughput analytical methods are compatible. In our case, we have an advantage that compounds are not bound to the surface and, therefore, MALDI-MS (very sensitive, fast and informative method) can be applied for the on-chip characterization. In the initial stages, we will of course utilize off-chip methods to characterize the synthesis efficiency, yields, limitations, etc.

Comment: A technical issue should be addressed. In the schemes, two chips are contacted through liquid-liquid contact. For this to be successful, there needs to be very good alignment of the two chips. Anything off even several microns (depends on spot size) will reduce effectiveness. How is this accounted for by the authors?

Response: Thanks for this comment. For the perfect aligning we have used an in-house developed aligner device. The procedure is discussed, and photographs of the device are added to the supplementary information.

The following figure was added to the supplementary information:

Supplementary Figure 4 | Alignment (“sandwiching”) device. (a) Photograph of the opened alignment device. (b) Photograph of the closed alignment device during the synthesis step. (c) Photograph of two sandwiched slides in the alignment device during the synthesis step. (d) Schematic describing the process of precise sandwiching of two droplet arrays. The alignment device consists of a lower and upper frame. Slide A is set into the lower frame of the alignment device and fixed with a spring holder. Slide B is set into the upper frame of the alignment device and fixed, too. The lower frame shows round rods in each edge of the frame, whereas the upper frame shows round recesses on the same positions. The upper frame is put on the lower frame so that the rods from the lower frame slide into the recesses of the upper frame and therefore fix both frames in the lateral position. The distance in height of both frames can be precisely controlled by four screws until the droplets of slide B merge with the droplets of slide A.

Comment: The ability to include biological reagents (such as plasmid DNA) is quite useful. The authors showed improved transfection, albeit with higher toxicity. This opens up opportunities for downstream applications, e.g., CRISPR. That may need to be addressed in more detail.

Response: We added a discussion of other potential biological applications (such as screening of membrane proteins and CRISPR) to the revised manuscript.

“The screening of lipidoids for their cell transfection efficiency using plasmid DNAs is an example of a biological assay that can be combined with synthesized libraries. Other possible biological experiments include, inter alia, CRISPR/Cas9-based gene editing, screening of membrane proteins, gene-knockdown, screening of cell spheroids or organoids.”

Reviewer 4:

Comment: The ms describes an essentially novel microarray droplet technology based on patterns with different types of wettabilities and chemical or biological compounds/cells allowing for the synthesis of model chemical compounds, converting them into an aqueous droplet array and finally testing a library of compounds in a cellular assay (here transfection).

The technology builds partly on microarray drop array techniques developed earlier in the KIT group, but the combination is entirely new in my view, addressing the highly important fabrication of chemical/pharmaceutical libraries and its evaluation in cell-based assays. This microarray microdroplet technique is highly innovative and has the potential of massively shortening the time and reducing the amount of compounds needed for library synthesis, microarray preparation, and cell-based highly parallel readouts.

The presentation of the result is very professional including state of the art, clear argumentation, background info, process development and readouts, nicely designed figures, and references.

A topic that is (I believe) not addressed is the issue of byproducts produced during the library synthesis of chemical compounds. Although turnover is quite impressive, small amounts of side products might have important undesirable effects such as cell toxicity. This should be addressed.

Response: We thank the reviewer for the positive response and appreciation. We agree with the reviewer that the on-chip parallel high-throughput purification is a very important but also interesting challenge that need to be better discussed in the manuscript. We are currently working on the implementation and optimization various strategies to achieve parallel high-throughput purification essential for such screenings. The advantages of our droplet microarray platform are that it is an open platform where all microcompartments are accessible externally, which potentially makes it compatible with parallel miniaturized and high-throughput extraction, crystallization, or even sublimation. We would like to emphasize that such purification methods are usually incompatible with most of the high-throughput miniaturized combinatorial platforms (droplet microfluidics, microfluidics in general, slipchip, etc.).

In order to address this point, we performed a proof-of-principle experiment showing the possibility to do on-chip purification by two-phase liquid-liquid extraction.

Comment: Also, the issue of what clean room standards would have to be used in a commercial fab lab would be useful to know.

Otherwise, I highly recommend publication after the proposed revisions. I believe that this publication will be of great interest to a large community in both academia and industry.

Response: We have given the following information in the method part about the precaution to be taken for fabrication:

“The on-chip synthetic procedure was performed in a standard chemistry fume hood.”

“The on-chip preparation of liposomes/lipoplexes was performed under sterile conditions using a standard sterile clean bench.”

“The on-chip reverse transfection was performed under sterile conditions using a standard sterile clean bench.”

REVIEWERS' COMMENTS:

Reviewer #1 (Remarks to the Author):

This reviewer is satisfied with the additional experiments performed by the authors in the revision and recommends that this manuscript be published in Nature Communications without revisions.

The authors clearly went to great lengths to address our concerns. The introduction is vastly improved and more relevant to the chemBIOS platform. The demonstration of compatibility of additional solvents in addition to DMSO, liquid-liquid extraction and combination with additional analytical techniques beyond MALDI adds to potential of the chemBIOS platform and greatly strengthens this manuscript. This reviewer agrees that some chemical reactions may be difficult to translate to this platform, but the work here is suitable as proof of concept. We will look forward to reading future studies expanding on the scope of the chemBIOS system and seeing applications using platform in other cell-based assays beyond transfection and with a variety of chemistry.

Reviewer #2 (Remarks to the Author):

Authors responded to all the remarks from referees, the revised manuscript is acceptable for publication.

Reviewer #3 (Remarks to the Author):

The authors have addressed the majority of my concerns and comments. One additional comment, however, needs to be made. In searching the literature, the droplet contacting system has actually been developed before. Lee et al. 2008, PNAS (<https://www.pnas.org/content/105/1/59>) showed human drug metabolism on a chip where human P450s were on one chip with gel spots and those were contacted with a second chip with human cells in culture in a spot. The two chips were aligned such that the spots from one chip can be in contact with the spots on the second chip. Obviously, the focus of the screening and synthesis in the current manuscript is distinct, but lack of referencing of a major paper (in PNAS) is regrettable.

Reviewer #4 (Remarks to the Author):

The ms has been carefully revised. I agree with all responses to questions/comments I raised, and consider also the responses to questions from other reviewers adequate and complete. I propose this ms to be accepted without further revisions.